# Organocatalytic asymmetric synthesis of *Si*-stereogenic silacycles

Jung Tae Han[1,2], Nobuya Tsuji [3], Hui Zhou [1], Markus Leutzsch [1] &
Benjamin List [1,3] ✉

A strong and confined Brønsted acid catalyzed enantioselective cyclization of bis(methallyl)silanes provides enantioenriched *Si*-stereogenic silacycles. High enantioselectivities of up to 96.5:3.5 er were obtained for a range of bis(-methallyl)silanes. NMR and ESI-MS studies reveal that the formation of a covalent adduct irreversibly inhibits turnover. Remarkably, we found that acetic acid as an additive promotes the collapse of this adduct, enabling full turnover. Experimental investigation and density functional theory (DFT) calculations were conducted to elucidate the origin of this phenomenon and the observed enantioselectivity.

Organosilanes are valuable substances in medicinal chemistry as well as in chemical synthesis and material science[1-5]. Silicon has several similarities with carbon (e.g., tetrahedral geometry and valency of four), but there are also distinct differences between the two elements (e.g., covalent radius, electronegativity and liphophilicity) making silicon an interesting (bio)isostere of carbon[4,5]. The synthesis of organosilicon compounds provides new opportunities to drug discovery and scent design. Among the diverse organosilicon compounds, *Si*-stereogenic silanes have recently demonstrated promising biological activities[6,7]. For example, Tomooka's group has shown that one *Si*-stereogenic silacyclopentane epimer displayed significant binding to the serotonin receptor whereas the opposite epimer showed no binding (Fig. 1a)[7]. These results underline the importance of silicon stereogenecity in biologically active molecules. However, methods for the preparation of organosilicon compounds possessing a *Si*-stereogenic center are much less developed as compared to carbon stereogenic molecules.

Even though advances have been made toward the synthesis of *Si*-stereogenic silanes over the past several decades, many of the reported syntheses have relied on stoichiometric chiral reagents or auxiliaries[8-15]. More recently, catalytic asymmetric variants including transition-metal catalyzed hydrosilylation of multiple bonds[16-25], alcoholysis of dihydrosilanes[26-28], Si−H bond insertion of carbenes[29,30], Si−C bond cleavage[31-35], C−H bond activation[36-43] and others[44-51] have been disclosed, but they have mostly depended on precious transition-metal catalysts, such as rhodium and palladium[52-56]. While several enzymatic

methods have also been developed[57,58], organocatalytic methods only recently emerged[59-63].

We recently reported a strongly acidic and confined imidodiphosphorimidate (IDPi) catalyzed synthesis of *Si*-stereogenic silyl ethers via desymmetrization of bis(methallyl)silanes (Fig. 1b)[61,64,65]. We also reported a dynamic kinetic asymmetric transformation (DYKAT) of racemic methallyl silanes[62]. During our previous desymmetrization study, we found that when 2,6-dimethylphenol was excluded, the corresponding silacycle was formed, albeit with poor yield and enantioselectivity (Fig. 1b). Based on this result, we speculated if we could develop the process to access enantioenriched *Si*-stereogenic silacycles. As the desired reaction product would still be an allylsilane, which is known to be activated by strong acids[66,67], a key challenge could be to identify a catalyst system that prevents further activation of the product. Herein, we report an IDPi catalyzed enantioselective cationic cyclization of bis(methallyl) silanes that provides access to *Si*-stereogenic silacycles (Fig. 1c). Critical to this discovery has been the identification of acetic acid as a catalyst turnover enabling additive. During the preparation of this manuscript, Yu and co-workers reported a chiral imidazolidinone catalyzed intramolecular aldolization of siladials to generate enal-containing *Si*-stereogenic silacycles (Fig. 1d)[63].

## Results

Initially, a series of chiral Brønsted acid catalysts were tested for the reaction of bis(methallyl)silane **1a** in toluene at −60 °C (Table 1).

[1]Max-Planck-Institut für Kohlenforschung, Kaiser-Wilhelm-Platz 1, 45470 Mülheim an der Ruhr, Germany. [2]Korea Institute of Science and Technology (KIST), Seoul 02792, Republic of Korea. [3]Institute for Chemical Reaction Design and Discovery (WPI-ICReDD), Hokkaido University, Sapporo 001-0021, Japan. ✉e-mail: list@kofo.mpg.de

## a. Biological activity of *Si*-stereogenic silacycle

$S_{Si}$ epimer
(biologically active)

$R_{Si}$ epimer

## b. Our previous work
*Organocatalytic synthesis of Si-stereogenic silyl ethers*

92% yield
97:3 er

*Initial finding of Si-stereogenic silacycle formation*

without
2,6-dimethylphenol

24% yield, 53:47 er

## c. This work
*Organocatalytic synthesis of Si-stereogenic silacycles*

AcOH additive

up to 96.5:3.5 er
(acid-sensitive allylsilane)

## d. Yu's work
*Organocatalytic synthesis of Si-stereogenic silacycles*

chiral
imidazolidinone

up to 98.5:1.5 er

**Fig. 1 | Biologically active *Si*-stereogenic silanes and organocatalytic synthesis of *Si*-stereogenic silanes. a** Biological activity of *Si*-stereogenic silacycles. **b** Our previous work; Organocatalytic synthesis of *Si*-stereogenic silyl ethers. **c** This work; organocatalytic synthesis of *Si*-stereogenic silacycles. **d** Yu's work; organocatalytic synthesis of *Si*-stereogenic silacycles.

While chiral phosphoric acid (CPA) **3**, disulfonimide (DSI) **4**[68,69] and imidodiphosphate (IDP) **5**[70] showed no reactivity, strongly acidic IDPi **6a** gave the desired product albeit in disappointingly low conversion and enantioselectivity (entries 1–4). We speculated the lower conversion of the cyclization reaction compared to our previous intermolecular reaction to be due to the formation of a covalent adduct. Accordingly, we expected that an achiral Brønsted acid additive could accelerate the reaction by efficiently liberating the free catalyst from the adduct. Indeed, phenol, benzoic acid and pivalic acid slightly enhanced the reactivities without change in enantioselectivities (entries 5–7). Acetic acid performed better than other screened additives (entry 8). With acetic acid, we further screened a range of IDPi catalysts. IDPi **6b** possessing 4-(*tert*-butyl)phenyl substituents at 3,3′-positions of the BINOL backbone slightly increased both reactivity and enantioselectivity (entry 9). Modifying the inner core of the IDPi from trifluoromethyl to perfluoronaphthalene-2-yl considerably improved the enantioselectivity as well as reactivity (entries 9–11). By lowering the reaction temperature to −100 °C, the enantioselectivity was further enhanced, albeit with modest conversion. The reactivity could be improved by increasing the amount of acetic acid (entries 12–14). In the absence of acetic acid, the reaction resulted in a disappointing conversion, indicating that an additive is necessary for the reaction (entry 15). Finally, slightly

higher catalyst loading (5 mol%) led to full conversion with high enantioselectivity (entry 16). The absolute configuration of the silacycle **2a** was determined by Mosher ester analysis after derivatization (see the Supplementary Information for details)[71,72].

With optimized conditions in hand, we next examined the substrate scope of the cyclization (Fig. 2). In general, reactions of bis(methallyl)silanes **1a**–**1g** with various alkyl substituents proceed to furnish the silacycle products in good yields and high enantioselectivities. The enantioselectivity decreased slightly as the steric hindrance of alkyl substituent increases from linear alkyl (**1a**–**1e**, 93:7 to 95:5 er), isopentyl (**1f**, 92:8 er) and isobutyl (**1g**, 91:9 er). The reaction of 2-naphthyl substituted silane **1h** afforded the desired products in good yield and high enantioselectivity. While substrate **1i** with *o*-methyl substituent yielded the product with drastically diminished enantioselectivity, substrates **1j** and **1k** with *m*- and *p*-methyl substituents exhibited high enantioselectivities. *p*-Methoxy and fluoro substituted silanes **1l** and **1m** were reactive, but slightly reduced enantioselectivities were observed. Of note, bis(methallyl) allylsilane **1n** and tris(methallyl)silane **1o** were feasible substrates for the reaction, leading to the corresponding silacycle products with high enantioselectivities. Finally, we found that cyclohexyl substituted silane **1p** showed low enantioselectivity compared to other silanes.

To gain mechanistic insight, we conducted several control experiments (Fig. 3). When enantioenriched silacycle **2a** was subjected to the reaction conditions in the presence of Tf-substituted catalyst **6b** which is more acidic than the corresponding perfluoronaphthyl-2-sulfonyl-substituted catalyst **6d**, silyl acetate **7** was formed as major product. In contrast, catalyst **6d** did not lead to side product **7**, indirectly suggesting that the stability of the silacycle under the reaction conditions depends on the acidity of the IDPi catalysts (Fig. 3a). However, despite the presence of less acidic catalyst **6d**, silacycle **2a** was completely transformed into side product **7** at room temperature, illustrating the temperature dependence of the silacycle stability (Fig. 3b). As expected, the less reactive bis(allyl)silane **1q** did not react under our reaction conditions (Fig. 3c).

To understand the role of the acetic acid additive, we conducted $^{31}P$ NMR spectroscopic studies with substrate **1a** and catalyst **6c** (Fig. 4). When 10 equiv. of silane **1a** was subjected to catalyst **6c** in toluene-$d_8$ solution, eight peaks were immediately observed in the $^{31}P$ NMR spectrum. Most likely, these peaks originated from in-situ generated covalent adduct **8** and Electrospray Ionization Mass Spectrometry (ESI-MS) analysis further supported the formation of this species[73]. Free catalyst **6c** was also observed with a significant *downfield* shift of phosphorous signal. Subsequent addition of 10 equiv. of acetic acid to the mixture led to complete regeneration of free catalyst **6c** and production of silyl acetate **7**. While this reaction reduces the overall yield of the desired product, it enables catalyst turnover.

Based on the control experiments and our NMR and MS studies, a catalytic cycle can be suggested (Fig. 5a). Protonation of the olefin provides ion-pair **I** consisting of a *β*-silicon stabilized carbocation and the IDPi anion. Cation-π cyclization of **I** leads to the formation of ion-pair **II**[74–76]. Intra ion pair proton transfer within **II** furnishes the *Si*-stereogenic silacycle **2**, regenerating free catalyst **6**. In parallel, ion-pair **II** can collapse to form covalent adduct **8** which is deleterious to the catalyst turnover. However, acetic acid liberates catalyst **6** from adduct **8**, enabling catalyst turnover.

To gain further insights into the detailed reaction mechanism and investigate the origin of the observed enantioselectivity, we conducted a computational study with substrate **1a** and catalyst **6d** (see the Supplementary Information for details). Calculation was performed at CPCM(Toluene)-ωB97M-V/(ma)-def2-TZVPP//r²SCAN-

**Table 1 | Reaction development[a]**

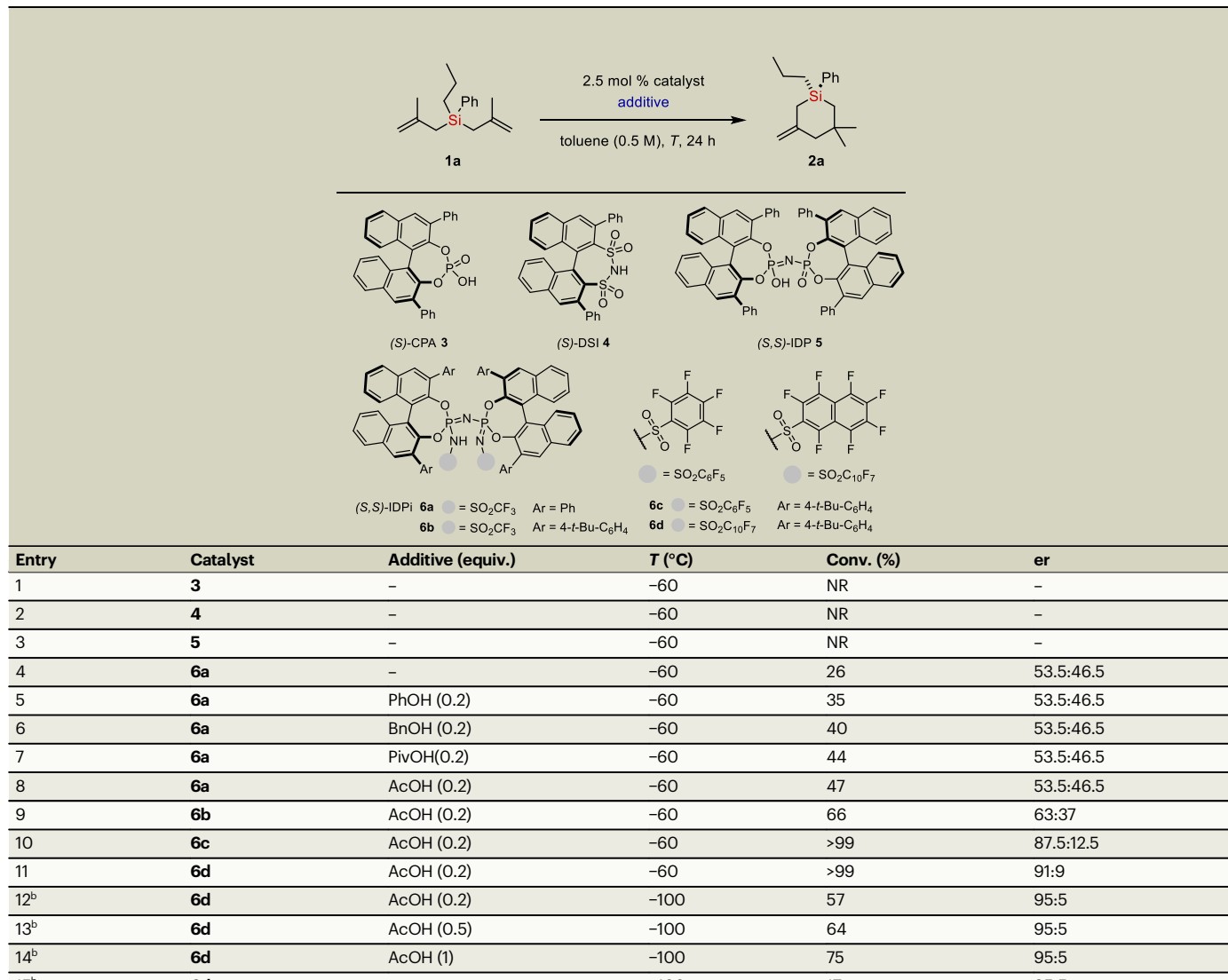

| Entry | Catalyst | Additive (equiv.) | T (°C) | Conv. (%) | er |
|---|---|---|---|---|---|
| 1 | **3** | – | −60 | NR | – |
| 2 | **4** | – | −60 | NR | – |
| 3 | **5** | – | −60 | NR | – |
| 4 | **6a** | – | −60 | 26 | 53.5:46.5 |
| 5 | **6a** | PhOH (0.2) | −60 | 35 | 53.5:46.5 |
| 6 | **6a** | BnOH (0.2) | −60 | 40 | 53.5:46.5 |
| 7 | **6a** | PivOH(0.2) | −60 | 44 | 53.5:46.5 |
| 8 | **6a** | AcOH (0.2) | −60 | 47 | 53.5:46.5 |
| 9 | **6b** | AcOH (0.2) | −60 | 66 | 63:37 |
| 10 | **6c** | AcOH (0.2) | −60 | >99 | 87.5:12.5 |
| 11 | **6d** | AcOH (0.2) | −60 | >99 | 91:9 |
| 12[b] | **6d** | AcOH (0.2) | −100 | 57 | 95:5 |
| 13[b] | **6d** | AcOH (0.5) | −100 | 64 | 95:5 |
| 14[b] | **6d** | AcOH (1) | −100 | 75 | 95:5 |
| 15[b] | **6d** | – | −100 | 17 | 95:5 |
| 16[b,c] | **6d** | AcOH (1) | −100 | >99 (70)[d] | 95:5 |

*NR* no reaction.

[a]Reactions were performed with substrate **1a** (0.025 mmol), catalyst **3–6** (2.5 mol %) in toluene (0.5 M); conversions (conv.) were determined by [1]H NMR analysis with dibromomethane as an internal standard; enantiomeric ratios (er) were measured by HPLC.

[b]The reaction was run for 7 days.

[c]5 mol % of catalyst was used.

[d]NMR yield; NMR yield was determined by [1]H NMR analysis with triphenylmethane as an internal standard.

3c level of theory to illustrate the relatively small energy difference between two enantiomers[77–79]. Protonation via **TS1** is suggested to be the enantiodetermining step of the catalytic cycle, as the subsequent C−C bond forming step is deemed facile. However, the energy difference between **TS1** and **TS1'**, which lead to enantiomers **I** and **I'**, is only 0.2 kcal/mol, suggesting that no significant enantioselectivity can be induced in this step (Fig. 5b). After careful analysis, we realized that deprotonation via **TS3** and covalent adduct formation via **TS4** are key to the observed high enantioselectivity. In the case of the major enantiomer, the deprotonation (**TS3**, −5.6 kcal/mol) is favored over the covalent adduct formation (**TS4**, −5.0 kcal/mol). In contrast, for the minor enantiomer, the latter (**TS4'**, −6.2 kcal/mol) is favored over the former (**TS3'**, −5.4 kcal/mol). Hence, it is likely that an enantioenrichment process via a parallel kinetic resolution (PKR) takes place at this stage[80,81]. The major factor here can be attributed

to the instability of **TS4** compared to **TS4'**, presumably due to steric hindrance induced by the substrate in **TS4** (Fig. 5c). Assuming that the transformation is clean enough to provide only the desired products (**2** and **2'**) and the adduct **7** (via **8** and **8'**), calculating the ratio between them based on the Boltzmann distribution provided the expected yield of the product as 59% and the calculated enantiomeric ratio as 95:5, which are in good agreement with the experimental results (**2a**, 65% yield, 95:5 e.r.) (see Supplementary Information for details). The proposed mechanism is also supported by the high enantioselectivity of product **2o** as the initial protonation intermediate is still achiral. Additionally, the energy of the covalent adduct **8** is found to be −21.4 kcal/mol (**8**) and −20.9 kcal/mol (**8'**), significantly lower than the intermediates in the catalytic cycle, suggesting that it cannot return to the catalytic cycle without an additive under the reaction conditions. Therefore, the addition of

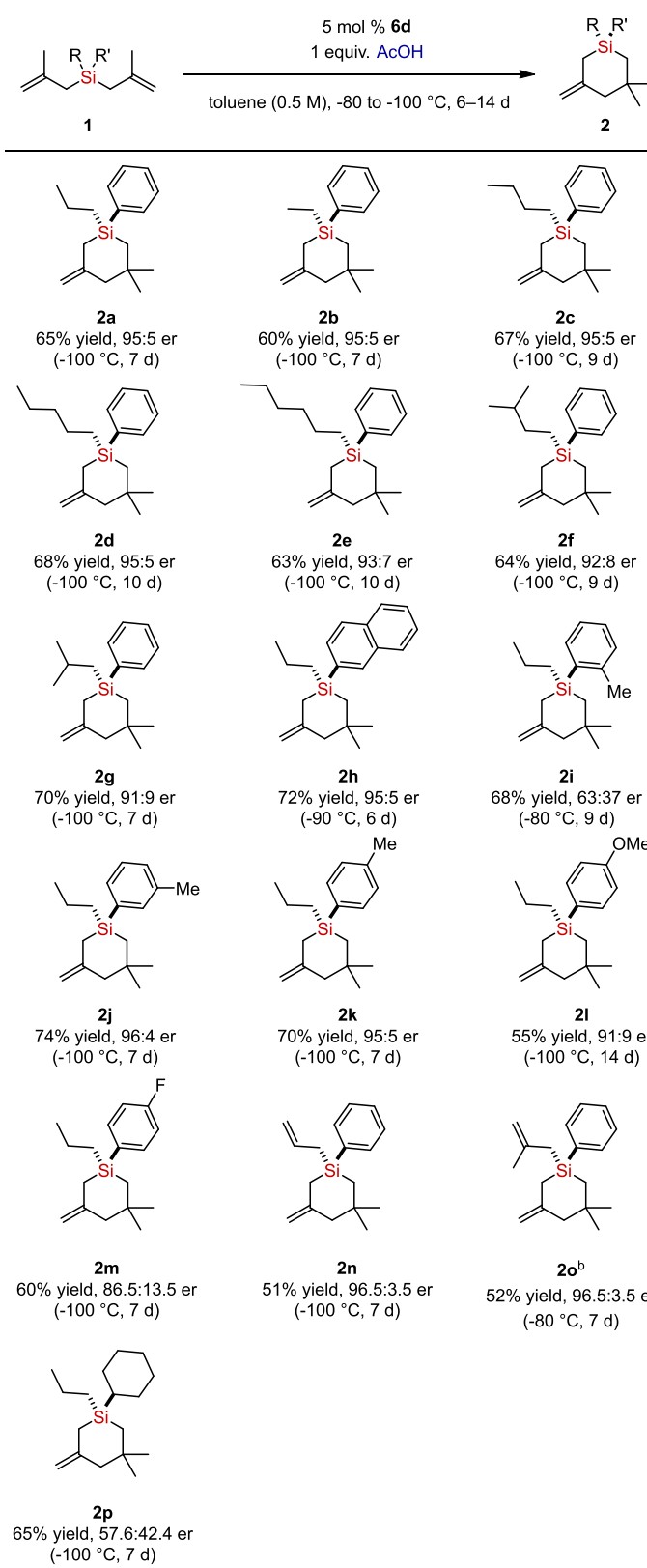

**Fig. 2 | Substrate scope of the reaction [a].** [a]Reactions were carried out with 0.1–0.2 mmol of substrate **1**, catalyst **6d** (5 mol %), acetic acid (1 equiv.) in toluene at the specified temperature for the specified time. Yields are for the isolated compounds. The enantiomeric ratios (er) were determined by HPLC analysis. [b]Pivalic acid was used instead of acetic acid.

**a** Effect of catalyst acidity on the stability of silacycle

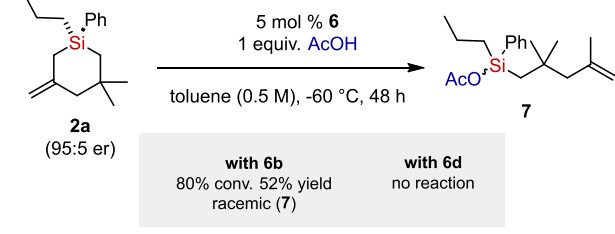

**b** Effect of temperature on the stability of silacycle

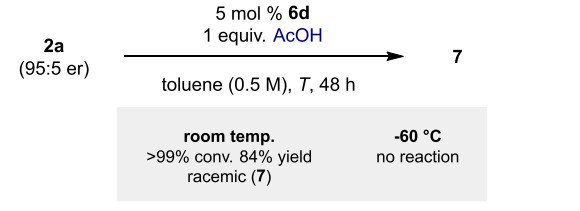

**c** Effect of allyl substituent on the reactivity

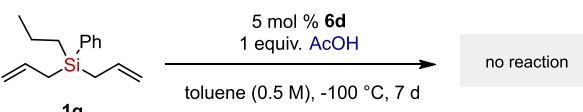

**Fig. 3 | Control experiments. a** Effect of catalyst acidity on the stability of silacycle. **b** Effect of temperature on the stability of silacycle. **c** Effect of allyl substituent on the reactivity.

acetic acid is crucial to achieve a high turnover in this transformation.

## Discussion

In summary, we have developed a strong and confined Brønsted acid catalyzed cyclization of bis(methallyl)silanes. A range of bis(methallyl)silanes were converted to the corresponding *Si*-stereogenic silacycles in a highly enantioselective manner. Mechanistic studies suggest formation of a covalent adduct limiting catalyst turnover. Addition of acetic acid facilitates the rapid regeneration of the catalyst from the covalent adduct, enabling turnover. Furthermore, computational studies suggest the protonation as the enantiodetermining step and the subsequent enantioenrichment as a key to achieve high enantioselectivity. Efforts to develop other organocatalytic methods for a range of *Si*-stereogenic silanes are currently underway.

## Methods

### General procedure for the enantioselective cyclization of bis(methallyl)silanes

A GC vial was charged with catalyst **6d** (5 mol %), toluene and acetic acid (1 equiv.), and the resulting mixture was cooled to −100 °C in a cryostat. After 10 min, bis(methallyl)silane **1** (0.1 or 0.2 mmol) was slowly added and the GC vial was stored for the given reaction time at the same temperature. After complete conversion indicated by TLC, the reaction was quenched with trimethylamine. The solvent was removed *in vacuo* and the mixture was purified by column chromatography on silica gel to afford the desired silacycle **2**.

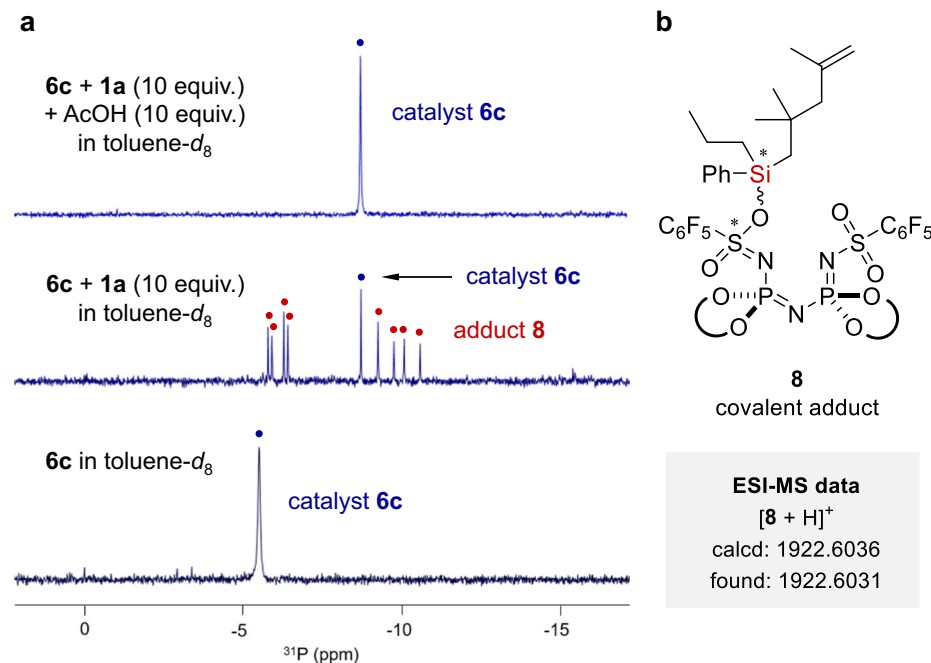

**Fig. 4 | Mechanistic studies. a** [31]P NMR experiments. **b** ESI-MS experiments.

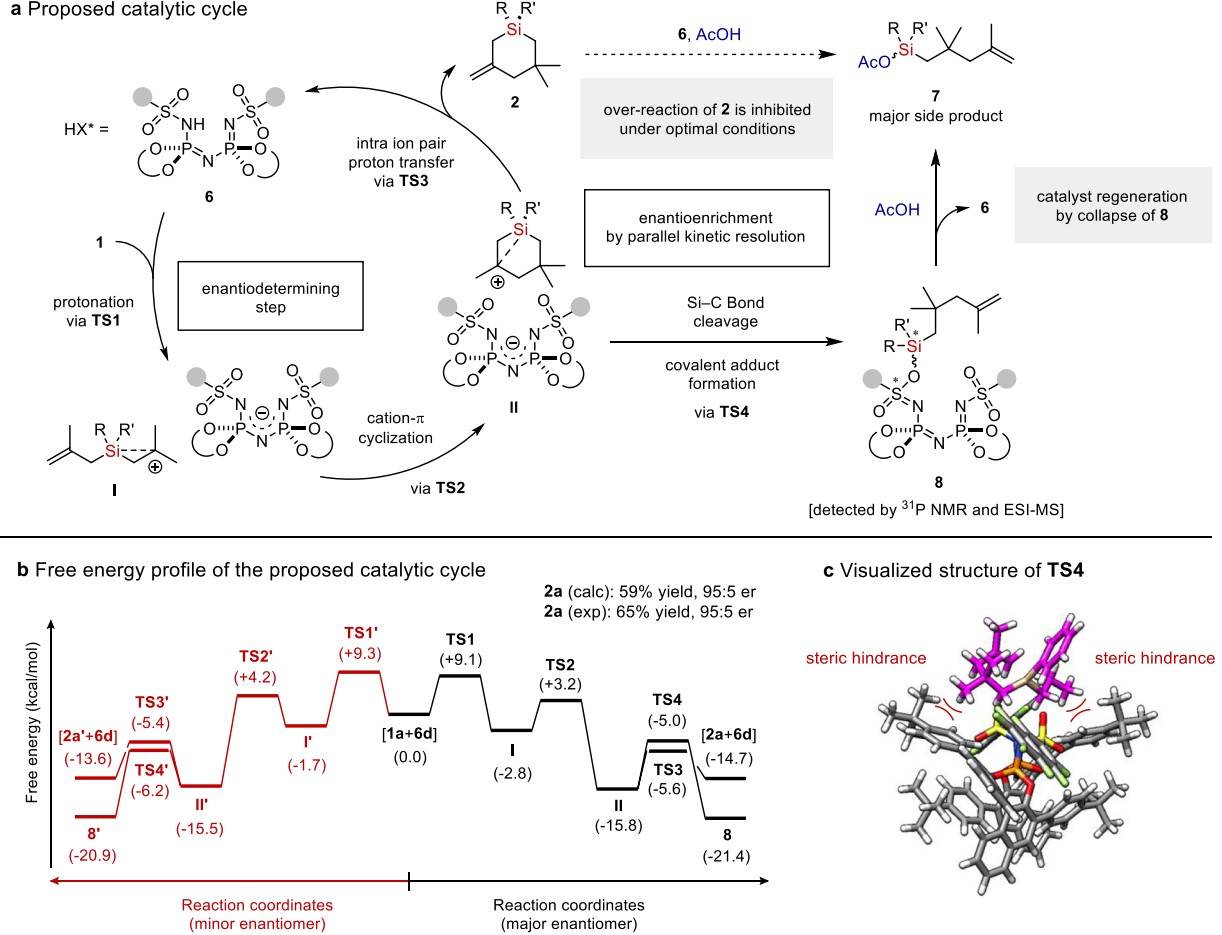

**Fig. 5 | Proposed catalytic cycle and computational studies. a** Proposed catalytic cycle. **b** Free energy profile of the catalytic cycle calculated at CPCM(Toluene)-ωB97M-V/(ma)-def2-TZVPP//r²SCAN-3c level of theory. The thermal corrections were calculated at 173.15 K. The energy profile leading to the major enantiomer is depicted in black, and the one to the minor enantiomer is in red. The calculated yield and enantiomeric ratio are given by the ratio between **2, 2′**, and **8 + 8′** based on the Boltzmann distribution (see the Supplementary Information for details). **c** Visualized structure of **TS4** leading to the covalent adduct **8**. Substrate is depicted in magenta.

## Data availability

The experimental procedures and analytical data supporting the findings of this study are available within the manuscript and its Supplementary Information files. Raw and unprocessed NMR data are available from the corresponding author upon request. Cartesian coordinates are available in Supplementary Data 1. Source data are provided with this paper.

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

## Acknowledgements

Generous support was received from the Max Planck Society, the Deutsche Forschungsgemeinschaft (Germany Research Foundation), the Leibniz Award (to B.L.), Germany's Excellence Strategy (Grant EXC 2033-390677874-RESOLV) and the European Research Council (Early-stage organocatalysis; to B.L.). This work was also financially supported by the Institute for Chemical Reaction Design and Discovery (ICReDD), which was established by the World Premier International Research Initiative (WPI), MEXT, JAPAN by List Sustainable Digital Transformation Catalyst Collaboration Research Platform offered by Hokkaido University, and by JSPS KAKENHI Grants 21H01925 and 20K22515. Part of the computation was performed using Research Center for Computational Science, Okazaki, Japan (Projects: 21-IMS-C303, 22-IMS-C129, and 23-IMS-C119). We also thank the technicians of our group and the members of our NMR, MS and chromatography groups for their excellent service.

## Author contributions

B.L. oversaw the project. J.T.H. developed the reaction and investigated the substrate scope, and conducted the mechanistic studies. J.T.H and M.L implemented NMR studies. N.T. performed the computational studies. Z.H. performed the computational studies on circular dichroism (CD). J.T.H. and B.L. wrote the manuscript.

## Funding

## Competing interests

The authors declare the following competing financial interest(s): a patent on the general catalyst class and its use in synthesis has been filed.
