## [Peer Review File · Nature Communications]

Organocatalytic asymmetric synthesis of Si-stereogenic silacyclesReviewers' Comments:

Reviewer #1:

Remarks to the Author:

In this manuscript, List and co-workers reported a strong and confined Brønsted acid catalyzed enantioselective cyclization of bis(methallyl)silanes, which provided enantioenriched Si-stereogenic silacycles with high enantioselectivities of up to 96.5:3.5 er. And the experimental results revealed that the formation of a covalent adduct irreversibly inhibits turnover and acetic acid as an additive promotes the collapse of this adduct, enabling full turnover. These findings were interesting and well-organized. Overall, this manuscript might be suitable for publication on Nature Communications after minor revision.

- (1) In Table 1, for the conversion of entries 10-11 and 16, the word "full" is not accurate, please use a data to support the perfect conversion. In fact, the isolated yield is not good, only 65% for 2a, why? The low yield might be due to the competitive reaction formed by side-product 7 after the use of acetic acid. What are the amounts of side-products in the screening of reaction conditions in Table 1?
- (2) The substrate scope is quite narrow and not suitable for aliphatic silanes. How about the reactivity for the heterocycle-substituted substrates in this reaction?

Reviewer #2:

Remarks to the Author:

This is an important and excellent article. Very minor fix needed:

Hindrance is spelled properly in the text, but is incorrectly hinderance in the computational figure.

Reviewer #3:

Remarks to the Author:

Organosilanes are a class of valuable compounds which have been widely used in fields of synthetic chemistry, medicinal chemistry and others. However, the methods for their asymmetric thesis containing Si-stereogenic centers remain far less developed. Particularly, taking advantage of organocatalytic methods are very rare. In this manuscript, List et al. developed a strong and confined Brønsted acid-catalyzed enantioselective cyclization of bis(methallyl)silanes, providing an intriguing method to access enantioenriched Si-stereogenic silacycles with moderate to good yields with generally good enantiocontrol. In these transformations, the additive-acetic acid has been found that it was critical to the success, and experimental investigation and DFT calculations were conducted and well elucidate the origin of this phenomenon and the observed enantioselectivity. After addressing the following issues. I think it would be suitable for published in Nature Communications.

- 1) The scope of the substrates is limited to bis(methallyl)silanes, other substituents rather than 2-methyl allylic group (or other substituents at the 2-position of the allylic group) are not suitable for this method (SI p40), the authors are expected to give a probable reason for this result. In addition, what's the result with the aryl bearing electron-withdrawing groups in 1.
- 2) The format of the superscript in Table 1 should be revised.

Reviewer #4:

Remarks to the Author:

The catalytic asymmetric synthesis of Si-stereogenic silanes has become a hot topic in recent years, and the enantioenriched Si-stereogenic silacycles have attracted increasing attention given that these non-natural chiral compounds can serve as versatile chiral building blocks, chiral reagents, and chiral auxiliaries. On the basis of their previous works, List et al. herein further reported a Brønsted acid catalyzed enantioselective intramolecular cyclization of bis(methallyl)silanes, which gave access to enantioenriched Si-stereogenic silacycles with good ee. The key point of this work is that they found acetic acid as an additive promotes the collapse of the adduct in the reaction, enabling full turnover. This is indeed an improvement for the IDPi catalyzed synthesis of Si-stereogenic silanes, however, the substrate scope is too special, and the fully carbon-substituted silacycle products barely display

downstream transformations. In addition, the reaction conditions are lack of efficiency (-80 to -100 C, 6-14 d). Although the reaction is interesting and appears to be useful in the synthesis of chiral silacycles, this manuscript does not include any particularly new information or application that could contribute to further progress in this area of research. Consequently, this reviewer concludes this work does not have the conceptual impact needed for publication in Nature Communications.

Suggestion: before the publication of this work in any journals, the authors should highlight other groups' works in this field in Fig 1, especially other organocatalytic methods towards Si-stereogenic silanes.

Reviewer #1 (Remarks to the Author):

(1) In Table 1, for the conversion of entries 10-11 and 16, the word “full” is not accurate, please use a data to support the perfect conversion. In fact, the isolated yield is not good, only 65% for 2a, why? The low yield might be due to the competitive reaction formed by side-product 7 after the use of acetic acid. What are the amounts of side-products in the screening of reaction conditions in Table 1?

Response: We now change the word “full” to “>99%”. We also provide the crude NMR of entry 16 (Table 1). The moderate yield of the desired product 2a (70% NMR yield) is indeed due to the formation of side product 7 (30% NMR yield). The NMR yield of entry 16 has been added to Table 1.

(2) The substrate scope is quite narrow and not suitable for aliphatic silanes. How about the reactivity for the heterocycle-substituted substrates in this reaction?

Response: 2-thiophene-substituted substrate was tested under the optimal reaction conditions, but the corresponding product was not formed. This result has been added to the supporting information as limitations of the method (see the supporting information, page S40).

Reviewer #2 (Remarks to the Author):

(1) Hindrance is spelled properly in the text, but is incorrectly hinderance in the computational figure.

Response: This mistake has been corrected in the main text and SI.

Reviewer #3 (Remarks to the Author):

(1) The scope of the substrates is limited to bis(methallyl)silanes, other substituents rather than 2-methyl allylic group (or other substituents at the 2-position of the allylic group) are not suitable for this method (SI p40), the authors are expected to give a probable reason for this result. In addition, what's the result with the aryl bearing electron-withdrawing groups in 1.

Response: The reactivity of the starting silanes is highly sensitive to the nature of allyl groups. The lack of reactivity of simple allyl and internal alkenyl groups correlates to their nucleophilicity (<https://www.cup.lmu.de/oc/mayr/reaktionsdatenbank2/fe/showclass/7>).

An aryl-substituted substrate containing the strongly electron-withdrawing CF₃ group was tested under the optimal reaction conditions, but the corresponding product was not formed. This result has been added to the supporting information as limitations of the method (see the supporting information, page S40).

(2) The format of the superscript in Table 1 should be revised.

Response: This mistake has been corrected.

Reviewer #4 (Remarks to the Author):

(1) however, the substrate scope is too special, and the fully carbon-substituted silacycle products barely display downstream transformations.

Response: Although we have tried to expand the scope of silane substrates over bis(methallyl)silanes, silanes below were unreactive under both optimal and modified conditions.

We described the derivatization of silacycle **2a** to a ketone and Mosher esters without a significant loss of enantiopurity (see the supporting information for details, page S18–21). Although the reaction

condition for each step has not been fully optimized, these results suggest synthetic utility of our products.

(2) In addition, the reaction conditions are lack of efficiency (-80 to -100 C, 6-14 d).

Response: The referee is correct in pointing out that our reactions, which have been optimized to feature high enantioselectivity, are rather slow. However, one should keep in mind that we are describing a novel organocatalytic approach to silacycles and are not suggesting this to be a commercial application. This is fundamental science and not yet a technology. Also, please note that our method is so far the only approach available and therefore defines the state of the art. There simply is no better or more efficient approach.

(3) Although the reaction is interesting and appears to be useful in the synthesis of chiral silacycles, this manuscript does not include any particularly new information or application that could contribute to further progress in this area of research. Consequently, this reviewer concludes this work does not have the conceptual impact needed for publication in Nature Communications.

Response: We respect this opinion but disagree. The transformation is novel and the first organocatalytic asymmetric approach to silacycles via cation π -cyclization.

(4) Suggestion: before the publication of this work in any journals, the authors should highlight other groups' works in this field in Fig 1, especially other organocatalytic methods towards Si-stereogenic silanes.

Response: Yu's organocatalytic method (chiral imidazolidinone catalyzed intramolecular aldolization of siladials) has now been added to Figure 1D.

Reviewers' Comments:

Reviewer #1:

Remarks to the Author:

The revised manuscript can be accepted without revision.

Reviewer #3:

Remarks to the Author:

After the author further experimental studies and explanation, I recommend publication of this manuscript in Nature Communications.

Reviewer #4:

Remarks to the Author:

In the revised manuscript the authors tried to address some of the comments raised by the reviewers, however, the reaction is still too specific with limited scope, as well as poor efficiency. The paper still falls short in this aspect. I therefore remain of the same opinion as before that this contribution is borderline for Nature Communications.

Reviewer #1 (Remarks to the Author):

The revised manuscript can be accepted without revision.

Reviewer #3 (Remarks to the Author):

After the author further experimental studies and explanation, I recommend publication of this manuscript in Nature Communications.

Response: The reactivity of the starting silanes is highly sensitive to the nature of allyl groups. The lack of reactivity of the simple allyl group correlates to their nucleophilicity (see Mayr's database of reactivity parameters for details;

<https://www.cup.lmu.de/oc/mayr/reaktionsdatenbank2/fe/showclass/7>).

Although the nucleophilicity of 2-phenylallylsilane ($N = 5.38$) is higher than methylallylsilane ($N = 4.41$) and allylsilane ($N = 1.68$), the starting silane **1r** was unreactive under both optimal and modified reaction conditions. These results indicate that the steric hindrance near the reaction site also correlates to the reactivity.

Unreactive substrates under both optimal and modified reaction conditions

optimal reaction conditions
5 mol % **6d**, 1 equiv. AcOH, toluene (0.5 M), $-100\text{ }^{\circ}\text{C}$, 7 d

modified reaction conditions
5 mol % **6e**, toluene (0.5 M), rt, 7 d

(S,S)-IDPi 6e

Ar = 3,5-(CF₃)₂-C₆H₃

Reactivity parameters

$N = 1.68$

$N = 4.41$

$N = 5.38$

$N = 0.65$

An aryl-substituted substrate containing the strongly electron-withdrawing CF₃ group was tested under the optimal reaction conditions, but the corresponding product was not formed. This result

has been added to the supporting information as limitations of the method (see the supporting information, page S40).

Reviewer #4 (Remarks to the Author):

In the revised manuscript the authors tried to address some of the comments raised by the reviewers, however, the reaction is still too specific with limited scope, as well as poor efficiency. The paper still falls short in this aspect. I therefore remain of the same opinion as before that this contribution is borderline for Nature Communications.